# Effect of Waste Clay Bricks on the Performance of Cemented Tailings Backfill and Its Damage Constitutive Model

Tianxiang Sun [1], Yifan Zhang [1], Kang Wang [1], Zhuoqun Yu [2]  and Yongyan Wang [1,*]

[1] College of Electromechanical Engineering, Qingdao University of Science and Technology, Songling Road No. 99, Qingdao 266061, China; sunt3483@gmail.com (T.S.); 17863978308@163.com (Y.Z.); 18254722266@163.com (K.W.)
[2] School of Civil Engineering, Yantai University, Qingquan Road No. 30, Yantai 264005, China; yzqun2007@126.com
[*] Correspondence: wangyongyan168@163.com

**Abstract:** In this study, two types of cemented tailings backfill (CTB) were prepared by breaking up the waste clay bricks (WCB) from dismantled buildings and replacing part of the cement with brick powder and part of the tailings with brick aggregate. The mechanical properties of the modified CTB were investigated. The results showed that the strength of CTB with 10% brick powder content increased by 16.24% at the curing age of 28 days, while the content of 20% brick powder only decreased by 0.11%. The strength of the brick aggregate with less than 20% content can be increased at each curing age, and the strength of the 15% brick aggregate at the curing age of 7 d and 28 d is increased by 65.54 and 58.8%. The failure strain of the CTB-containing brick powder decreases with the extension of the curing time, and the failure strain of the CTB-containing brick aggregate decreases significantly at the curing age of 7 days. In addition, a three-stage damage constitutive model was established based on the results obtained in the present study, and the laboratory test results of CTB under uniaxial compression were effectively described with high confidence.

**Keywords:** cemented tailings backfill; waste clay bricks; unconfined compressive strength; failure strain; damage constitutive model

## 1. Introduction

As demand for mineral resources in China rises dramatically and shallow surface mineral resources are depleted, mineral development will increase, and deep mining will become the norm [1]. The continuous deepening of mineral extraction and the void areas left behind pose serious safety hazards, and the surface piles of tailings also risk inducing cave-ins, landslides, and collapse [2]. The filling mining method can effectively solve these problems and increase the recovery rate of mineral resources [3–5]. Cement tailings backfill (CTB) is a multi-phase material formed by mixing a certain proportion of tailings and water with cement as the main cementitious material through mixing, shaking, curing, etc. It is used for underground mine filling and has the advantages of no pollution, low energy consumption and good roof connection performance [6–8]. The mechanical properties of the CTB are an essential basis for assessing the safety of mining operations and for calculating the occurrence of geological hazards. Most of the CTB materials in a quarry are subjected to compression and may be exposed to impact stresses and high strains from the surrounding rock [9]. The mechanical properties of CTB in compression are largely governed by its stress-strain relationship in compression [10].

Moreover, the damage constitutive model of CTB is very effective in evaluating and predicting its mechanical properties under compression. Many scholars have extensively investigated damage constitutive models for CTB [11–13]. The damage constitutive model of CTB is a Scientific theory for safe mining and disaster prevention of underground

orebodies. However, different materials and mixture proportions will also make the Stress–strain curve of CTB under compression very different. In some cases, the development trend of the Stress–strain curve may not apply to other models, so it is essential to propose corresponding damage constitutive models for experimental data of different materials and proportions.

With the increasing global attention to sustainable development and environmental protection, the demand for reducing resource consumption and waste generation is also becoming increasingly urgent. Therefore, using recycled waste in building materials has received widespread attention and research from scholars. Acchar et al. [14] found that applying clean Construction waste with a content of less than 20% in the production of building ceramics will not cause significant changes in performance. The research of Kusiorowski et al. [15] indicates that adding slag and ash enhances the sintering process of clinker ceramics and can be considered a secondary raw material for producing ceramic clinker products. Lemougna et al. [16] used waste Glass wool and Spodumene tailings to produce building ceramics and found that adding ground Glass wool waste can enhance the densification and strength development of building ceramics at lower temperatures. Beskopylny et al. [17] and Shcherban et al. [18] used the seed shell of the rubber tree and coconut fibers, two agricultural wastes, to apply in concrete materials, and found that these two wastes can not only be cost-effective but also improve the mechanical properties of concrete materials. Yilmaz et al. [19] and Chen et al. [20] studied the feasibility of removing Construction waste in CTB. They found that, at an appropriate content, the removed Construction waste can improve the internal structure of Construction waste and increase the strength of Construction waste. In addition, the application of waste materials with pozzolanic characteristics to construction production has also become a method of great concern. Many scholars have conducted extensive research on the impact of waste glass on the performance of concrete, and numerous studies have shown that waste glass has great application potential and many benefits in the field of concrete [21–25]. Karalar et al. [26] found that adding coal bottom ash can alter the bending and fracture behavior of reinforced concrete beams. Cheng et al. [27] found that applying fly ash to backfill materials can reduce drying shrinkage. Zhao et al. [28] used activated rice husk ash and ground granulated blast furnace slag to prepare cementitious materials for cement slurry backfill and found that 10% rice husk ash can achieve higher strength. Yue [29] and Zhang et al. [30] found that waste glass can replace some cement or be used as a binder for CTB with low-purity metakaolin, which can reduce backfilling costs.

Waste clay brick (WCB) is also a recyclable material with pozzolanic characteristics. WCB usually comes from defective bricks produced while manufacturing clay bricks or demolished buildings. In contrast, many abandoned clay bricks are produced in China from demolished buildings. In traditional Chinese architecture, brick and stone structures were commonly used, especially in residential construction. With the rapid development of urbanization and higher building requirements in China in recent decades, many traditional buildings, mainly clay bricks, have been demolished or abandoned as Construction waste due to damage, aging, design changes, or no longer meeting building requirements. According to literature reports, WCB may account for 50%–70% of China's Construction waste production [31]. In addition, with the popularity of concrete as the main building material and the inability of some bricks to be effectively recycled or reused in the demolition or reconstruction process, most of the past clay brick-based buildings will eventually become Construction waste. According to statistics, in the past 60 years, China has produced at least 20 billion cubic meters of clay bricks [32]. In the future, if these abandoned clay bricks, which have been turned into Construction waste, are not recycled, they may adversely affect the environment and land space. The material used to manufacture clay bricks is natural clay, consisting mainly of $SiO_2$ and $Al_2O_3$, which has the pozzolanic reaction and can react with $Ca(OH)_2$ [33–35].

Among other things, some researchers have reported on the effect of adding brick powder or brick aggregates to the mechanical properties of concrete. Ortega et al. [36]

found that concrete with up to 20% cement content replaced by waste brick powder had good serviceability in the long term. Letelier et al. [37] found that replacing natural aggregate concrete with waste brick powder at 5%, 10% and 15% of the cement content did not significantly alter the original compressive strength. Dang et al. [38] found that the compressive strength of WCB as fine aggregate was higher than plain concrete at both 25% and 50% replacement, other variables being equal. Zhao et al. [39] found that when the particle size of the waste brick powder is smaller, it not only accelerates the early cement and shortens the setting time but also results in faster growth of compressive strength and a denser microstructure. Currently, most tailings backfill techniques use cement as a binder; the cost of cement accounts for approximately 60% to 80% of the cost of the backfill material [40]. In recent research, Ercikdi et al. [41] applied WCB generated in production to sulfide-rich tailings and found that replacing cement improved strength and sulfate resistance while reducing filling costs. Another disadvantage of tailings is that the aggregate is too fine. Discarded clay bricks can bring in large-diameter aggregates in the utilisation process, and adding coarse aggregates can improve compressive strength [42]. The waste brick powder can bring both a cheap cementing agent and a large aggregate addition, so it has great potential for use in mining backfill. In addition, due to the wider particle distribution and different and complex chemical composition in the tailings in CTB, these factors often lead to significant differences in mechanical behavior. Therefore, it is necessary to conduct targeted mechanical research on WCB in CTB containing low sulfide tailings to have a reasonable and effective theoretical basis in backfill mining.

This study divides the demolished WCB into two materials, brick powder and brick aggregate, by particle size crushing and screening and applies them to CTB materials. CTB specimens with brick powder and CTB specimens with brick aggregate were prepared by varying the ratio of brick powder to cement and the ratio of brick aggregate to tailings to reduce backfill costs and improve compressive strength in CTB applications. Lab tests such as slump, bleeding rate, unconfined compression, scanning electron microscopy (SEM) and X-ray diffraction (XRD) were carried out. In addition, based on the stress-strain results of the lab tests, a three-stage damage constitutive model was proposed for the compaction stage, linear elastic stage, and post-peak failure stage. Finally, the CTB damage constitutive model was studied and validated.

## 2. Materials and Methods

### 2.1. Raw Materials

The main raw materials used in this experiment include tailings, ordinary Portland cement (No. 42.5), tap water and waste clay bricks. The tailings originated from an iron mine in Jining City, Shandong Province, China, as shown in Figure 1a. Provided by Pengsheng Maike Municipal Engineering Construction Co., Ltd. in Qingdao, Shandong Province, China, WCB conducted sampling at four corners of its construction site. All WCB comes from building demolition, as shown in Figure 1b. After the WCB was crushed, a portion of the brick powder below 80 mesh was screened out to replace part of the cement, and the remaining unscreened portion was used as brick aggregate to replace part of the tailings. The sample of the brick powder and brick aggregate are shown in Figure 1c,d.

Laser particle size analysis of the tailings and brick aggregate was carried out using a Malvern 2000 laser particle size analyser, and the particle size distributions determined are shown in Figure 2a. The cumulative particle size distribution of the tailings was analysed as $d_{10} = 4.703$ μm, $d_{30} = 70.963$ μm, $d_{50} = 183.865$ μm, $d_{60} = 244.016$ μm, coefficient of nonuniformity $C_u = 51.89$ and coefficient of curvature $C_c = 4.39$. The cumulative particle size distribution of the brick aggregate was analysed as $d_{10} = 18.666$ μm, $d_{30} = 56.368$ μm, $d_{50} = 142.804$ μm, $d_{60} = 224.403$ μm, coefficient of nonuniformity $C_u = 12.02$ and coefficient of curvature $C_c = 0.76$. Both aggregates are poorly gradation and have a large number of coarse particles. The chemical composition of tailings, WCB, and cement was determined by X-ray fluorescence spectroscopy (XRF) using the Zenium equipment of PANalytical B.V. through powder compression. The results are shown in Table 1. The XRD patterns of

tailings and WCB are shown in Figure 2b,c. It can be found that the main mineral components of the tailings are Quartz, Anthophylite, Hematite, Calcite and Clinochlore, while the main mineral components of WCB are Quartz, Illite, Albit and Hematite. The above three analysis methods were completed by the School of Materials, Qingdao University of Science and Technology staff.

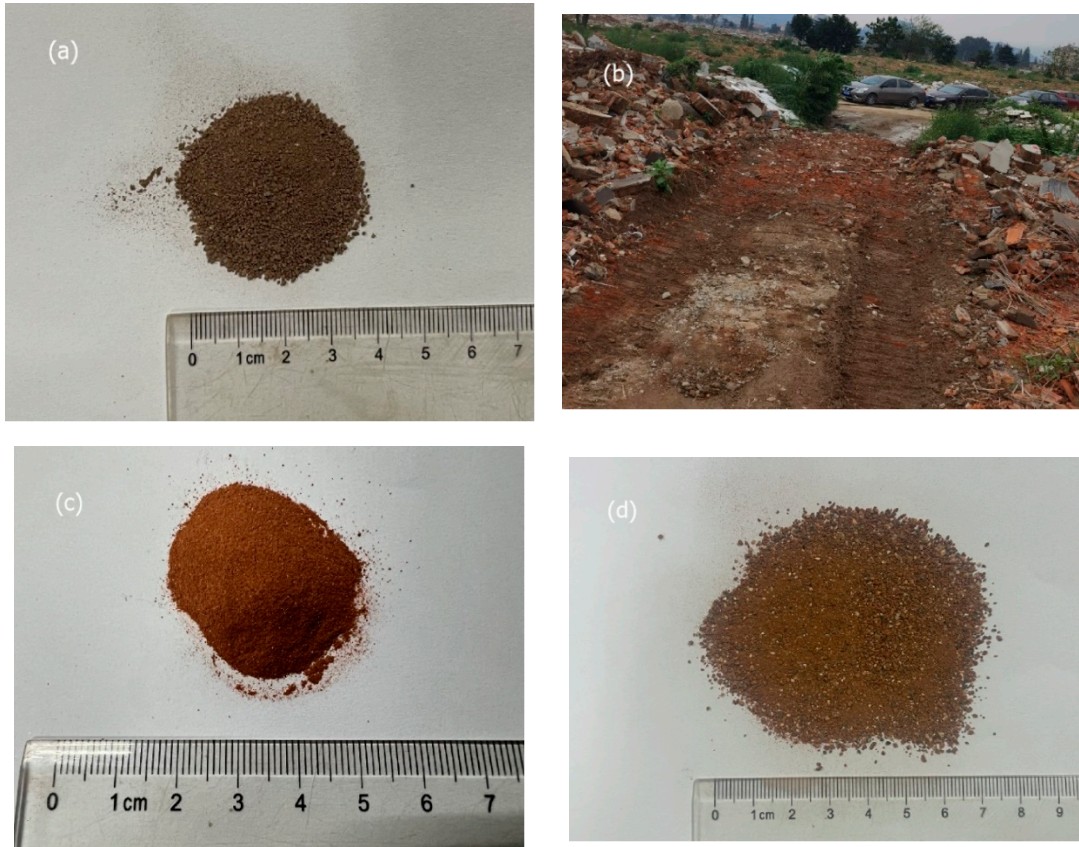

**Figure 1.** (**a**) Tailings sample; (**b**) Sampling site; (**c**) Brick powder sample; (**d**) Brick aggregate sample.

**Table 1.** Chemical composition of tailings, WCB, cement.

| Material | Tailings | WCB | Cement |
|:---:|:---:|:---:|:---:|
| $SiO_2$ | 27.59 | 64.09 | 21.40 |
| $Al_2O_3$ | 10.50 | 19.57 | 4.31 |
| $Fe_2O_3$ | 24.53 | 6.96 | 4.91 |
| CaO | 17.99 | 1.87 | 62.34 |
| MgO | 13.93 | 1.35 | 3.00 |
| $SO_3$ | 1.37 | 0.15 | 2.20 |
| $Na_2O$ | 1.32 | 1.56 | - |
| $K_2O$ | 0.82 | 3.11 | - |
| $TiO_2$ | 0.63 | 0.94 | - |
| MnO | 0.30 | 0.12 | - |
| $P_2O_5$ | 0.19 | 0.12 | - |

(a)

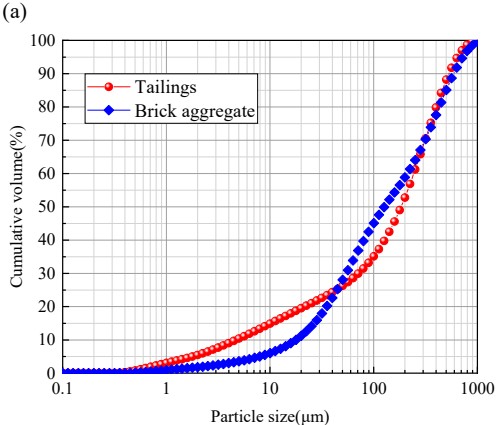

(b)

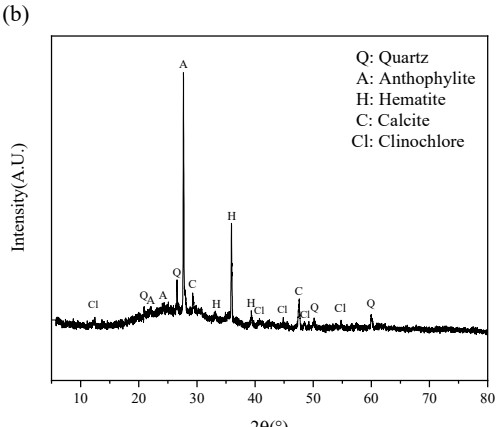

(c)

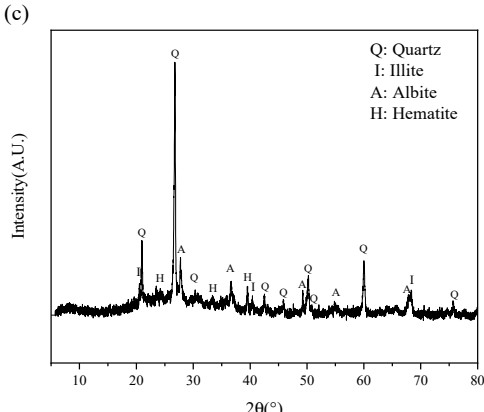

**Figure 2.** (**a**) Particle size distribution of tailings and brick aggregate; (**b**) XRD pattern of tailings; (**c**) XRD pattern of WCB.

### 2.2. Preparation of Specimens

Before preparing the CTB specimens, the tailings and brick aggregate were placed in an electric blast drying oven at 105 °C and dried for 24 h. After they had cooled naturally, CTB specimens with different contents of brick powder (10%, 20%, 30%, 40%) and brick aggregate (5%, 10%, 15%, 20%, 25%) were prepared. CTB should achieve a strength of at least 0.7 or more at 28 days [43]. This experiment sets the ash sand ratio (mass ratio of binder to aggregate) as 1:8, and the solid mass concentration is 75%. The water and solids mixture was stirred for 10 min to produce a homogeneous slurry, which was then slowly poured into a cylindrical mould of 50 mm diameter and 100 mm height and allowed to settle naturally.

The test was carried out for a consolidation time of 12 h. After reaching the consolidation time, the specimens were demoulded and placed in a constant temperature and humidity chamber at 20 °C and 95% humidity for 3 d, 7 d and 28 d. A simple format code was used to represent the different compositions of the CTB specimens. The alphabets 'C', 'T', 'P' and 'A' denote cement, tailings, brick powder and brick aggregate, respectively. For example, "CTP10" indicates a CTB specimen with 10% brick powder and "CTA5" indicates a CTB specimen with 5% brick aggregate. The names and proportioning information of CTB specimens are shown in Table 2. To ensure reproducible results, at least two specimens of each specimen were prepared for subsequent experimental testing.

**Table 2.** The name and proportioning information of the CTB specimen.

| Number of Specimens | Binders | | Aggregate | |
|---|---|---|---|---|
| | Cement (%) | Brick Powder (%) | Tallings (%) | Brick Aggregate (%) |
| $CTP_0(CTA_0)$ | 100 | 0 | 100 | 0 |
| $CTP_{10}$ | 90 | 10 | 100 | 0 |
| $CTP_{20}$ | 80 | 20 | 100 | 0 |
| $CTP_{30}$ | 70 | 30 | 100 | 0 |
| $CTP_{40}$ | 60 | 40 | 100 | 0 |
| $CTA_5$ | 100 | 0 | 95 | 5 |
| $CTA_{10}$ | 100 | 0 | 90 | 10 |
| $CTA_{15}$ | 100 | 0 | 85 | 15 |
| $CTA_{20}$ | 100 | 0 | 80 | 20 |

*2.3. Test Method*

The slump test is a simple and quick test method that many academics use to measure the flowability and transportability of CTB. The slump test is carried out in accordance with ASTM C 143 to assess the workability of the filler at 75% solids. The slump is calculated for each mixture to an accuracy of 5 mm.

The bleeding rate is an important parameter reflecting the shrinkage characteristics of CTB and the effectiveness of capping in the quarry. The lower the bleeding rate, the lower the CTB shrinkage rate and the better the capping effect. To test the water secretion rate, 800 g of each material required for the filling is poured into a 1000 mL measuring cup and stirred with a glass rod until the paste is homogeneous and no air bubbles are generated on the upper surface. Then let the slurry stand for 5 min and suck out the water from the upper layer with a pipette until there is no water on the upper surface of the slurry. Finally, the water secretion rate is calculated by the percentage of the total amount of water sucked out by the suction tube and the total amount of water mixed into the slurry.

The unconfined compression test is a common test method used to analyse the mechanical properties of CTB. The conditioned specimen is placed on the steel platform of the test machine, and the indenter of the test machine is controlled to be in contact with the upper surface of the specimen. The unconfined compression test is then carried out with a constant displacement loading mode at a rate of 0.2 mm/min, and the curve data is started and recorded until the loading is stopped when the specimen shows strong macroscopic damage.

The preparation and testing method of the specimen is shown in Figure 3. In addition, field emission gun scanning electron microscope (Hitachi SU8000, Hitachi High-Tech, Tokyo, Japan) and X-ray diffractometer (Riga Library MiniFlex 600, Rigaku, Tokyo, Japan) was used to conduct scanning electron microscope (SEM) analysis and X-ray Diffractometer (XRD) analysis on the microstructure of brick powder, brick aggregate and CTB samples.

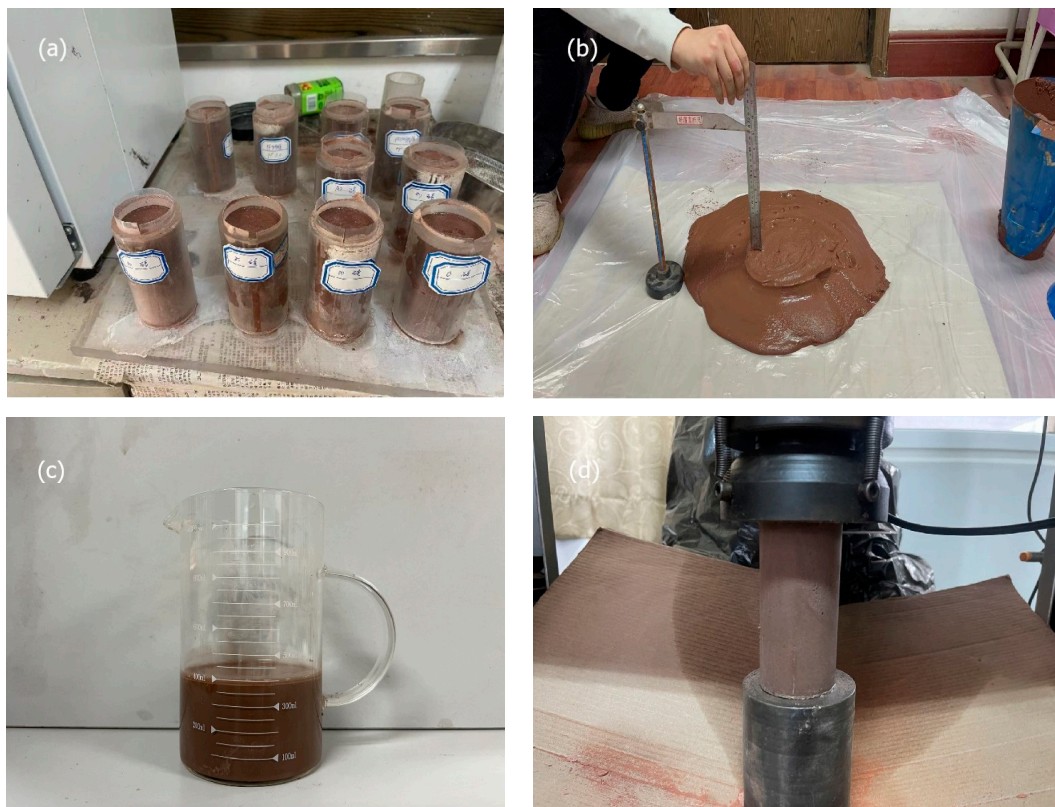

**Figure 3.** (**a**) Preparation of specimen; (**b**) Slump test; (**c**) Bleeding rate test; (**d**) Unconfined compression test.

## 3. Results and Discussion

### 3.1. Slump Value

The lab test results of the slump value and the bleeding rate of CTP and CTA are shown in Figures 4 and 5. From Figure 4, it can be seen that the slump value of CTB slurry remains unchanged within 20% of the brick powder content at 230 mm. When the brick powder content exceeds 20%, it decreases with the increase of the brick powder ratio. This phenomenon is consistent with the results reported by Bayram et al. [41]. From Figure 4, it can be seen that the slump value of CTB slurry decreases with the increase in brick aggregate ratio. This is mainly due to the high-water absorption of WCB. According to the Chinese national standard (GB/T 39489-2020), the slump value requirement is between 18 and 26 cm, and it is clear from the lab test results that all the CTBs meet the requirements for pipeline transport and flow properties [44].

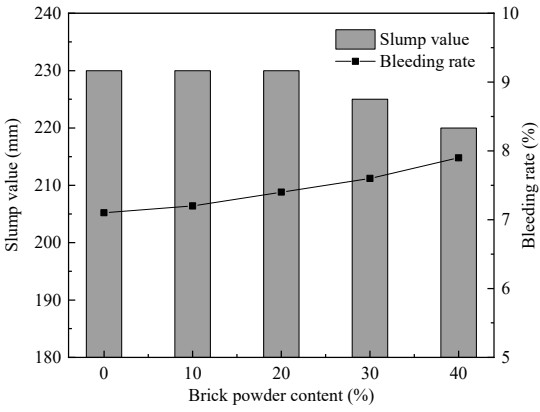

**Figure 4.** The slump value and the bleeding rate of CTP.

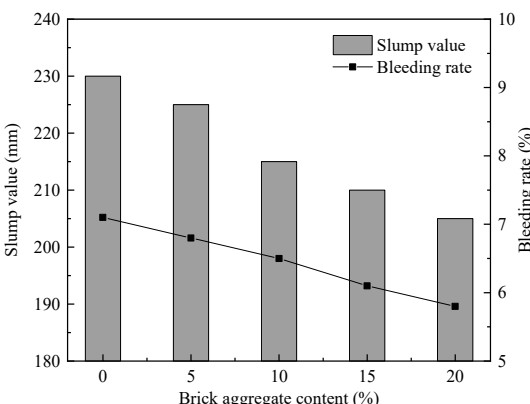

**Figure 5.** The slump value and the bleeding rate of CTA.

*3.2. Bleeding Rate*

It can also be seen from Figure 5 that the bleeding rate of the CTB slurry decreases with an increase in the proportion of brick aggregate. This shows that adding brick aggregate makes the CTB slurry more water-retentive due to its water absorption when the cement content is the same. It can also be seen from Figure 4 that with the increase in the proportion of brick powder, the bleeding rate of CTB slurry increases. In addition, the lower the cement content in a CTB slurry, the more bleeding rate there is. This indicates that the high-water absorption of the brick powder replacing the cement does not achieve the water retention properties of the cement. This may be because the fine particles in the brick powder fill the tiny pores between the tailings and the cement, making the CTB denser and displacing the water in it [45].

*3.3. Unconfined Compressive Strength (UCS)*

The unconfined compressive strength (UCS) of CTP at different curing ages is shown in Figure 6. It can be seen that the more the brick powder content, the smaller the UCS at the curing age of 3 d. At the curing age of 7 d and 28 d, the UCS tends to increase and then decrease with the increase of brick powder content, and the highest value of UCS in both ages is $CTP_{10}$, which is 1.14 MPa and 2.05 MPa, respectively. This indicates that the strength enhancement effect of brick powder for the curing age 3 d is slight, and its strength is mainly derived from the cement content and the hydration reaction of the cement, which is also insufficient, with fewer hydration products and lower strength. However, the hydration reaction gradually increases with the extension of the curing age, the $Ca(OH)_2$ in the hydration products, and the potential activity of the brick powder is fully stimulated, in which the active $SiO_2$ and $Al_2O_3$ will consume the $Ca(OH)_2$ in the hydration products of the cement for a secondary hydration reaction, which has a significant enhancement effect on the strength at the curing age of 7 d and 28 d. It can also be found that at 7 d of curing age, the UCS of $CTP_{10}$ is 1.6% higher than that of $CTP_0$, at 28 d of curing age, the UCS of $CTP_{10}$ is 16.24% higher than that of $CTP_0$, and the UCS of $CTP_{20}$ is only 0.11% lower than that of $CTP_0$. This indicates that up to 10% brick powder can increase the UCS of the original CTB at longer curing ages, and up to 20% brick powder content of CTB can maintain the UCS of the original CTB. Kulekci et al. [46] also obtained similar results, but its CTB had a higher UCS enhancement effect. This is because the addition of brick powder weakens the negative impact of sulfate on strength.

The unconfined compressive strength of CTA at different curing ages is shown in Figure 7. It can be seen that the higher the brick aggregate content at the curing age of 3 d, the larger the UCS. At the curing age of 7 d and 28 d, the UCS tends to increase and decrease as the brick aggregate content increases, with $CTA_{15}$ having the highest UCS of 1.86 MPa and 2.81 MPa, respectively. This indicates that the higher the brick aggregate content at 3 d of the curing age, the finer particles in the brick aggregate where the pozzolanic reaction occurs and the more hydration products. However, at the curing age of 7 d and 28 d, $CTA_{20}$,

due to the high-water absorption of brick aggregate caused by the free water inside CTB, is adsorbed by a large amount of brick aggregate so that its internal free water content decreases with the increase of brick aggregate. The free water may have been consumed in large quantities at 3 d of the curing age, resulting in insufficient hydration reaction and slow reaction speed of subsequent cement, while the hydration reaction of cement and the pozzolanic reaction of fine particles in the brick aggregate are mutually promoting and mutually restricting processes. The inadequacy and slow hydration reaction rate will, in turn, slow down the rate of the pozzolanic reaction, resulting in a slowdown in the intensity growth rate in 7 d and 28 d of curing age. It is also possible that the water adsorbed by the brick aggregate will enter the microfractures and microporosity under pressure and accelerate the formation of cracks, leading to a reduction in structural strength due to the increased moisture content of $CTA_{20}$. A higher UCS growth rate for CTB from 0% to 15% brick aggregate content can be found in $CPA_5$ compared to $CPA_0$ at the curing age of 28 d. This also reflects the faster rate of $CPA_5$ hydration reactions and the pozzolanic reaction, which produces hydration products. In addition, the CTB with brick aggregate had the highest strength growth rate of 51.68% at 3 d of curing age, 65.54% at 7 d of curing age and 58.8% at 28 d of curing age compared to $CPA_0$ alone. This contrasts CTB containing brick powder, where a suitable brick aggregate content can give CTB a strength advantage at curing ages 3 d and 7 d.

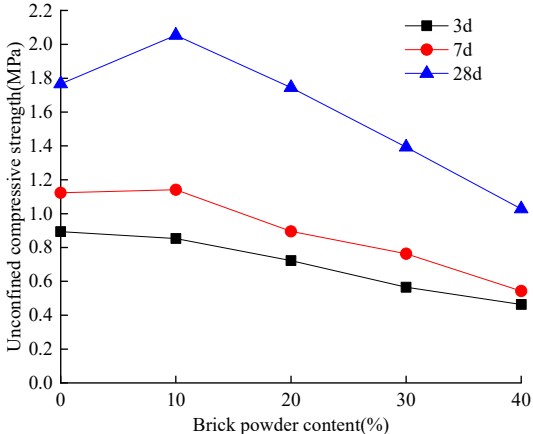

**Figure 6.** The unconfined compressive strength of CTP at different curing ages.

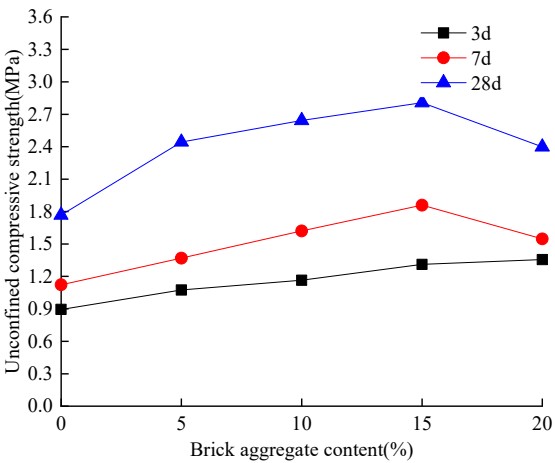

**Figure 7.** The unconfined compressive strength of CTA at different curing ages.

From the selling price of cement plants and construction companies, the price of cement is about RMB 400/t and after crushing and processing, the WCB is about RMB

100/t after processing. If the sole consideration is maintaining strength in the later stages, replacing cement with an appropriate level of brick powder will reduce the cost of backfilling. Replacing tailings with brick aggregate will increase the strength of the CTB at all curing ages. Although it will raise some of the cost of backfilling, there are economic incentives related to construction waste recycling in various Chinese provinces and cities. The environmental benefits of reducing carbon emissions from cement and mitigating pollution from construction waste ground accumulation cannot be ignored.

*3.4. Failure Strain*

Failure strain refers to the peak stress axial strain, which fully reflects the deformation characteristics of the CTB specimen at ultimate strength conditions. Figure 8 shows the failure strain of CTP at different curing ages. It can be seen that the CTB specimens containing brick powder have a significant tendency to decrease in failure strain as the age of curing increases, but the $CTP_0$ does not vary much from one curing age to another. The reduction in pore space within the CTB during the curing process can be inferred from the increase in the bleeding rate of the CTB containing brick powder. The production of AFt is affected by the number and size of the pores in the internal structure of the CTB. In addition, as the pozzolanic reaction, $Ca(OH)_2$ is consumed in the specimen [47], and the amount of AFt produced is affected. The AFt crystals grow anisotropically and are needle-like so that at high strains, CTB specimens containing brick powder may not crack quickly enough to prevent rapid failure due to the reduced number of AFt crystals produced and the dispersion of internal stresses when compressed. In addition, another reason for the decrease in peak strain is that the interior of CTB containing brick powder is more compact, and the hydration products of the pozzolanic reaction fill the internal pores. Others have also obtained similar results [41,46].

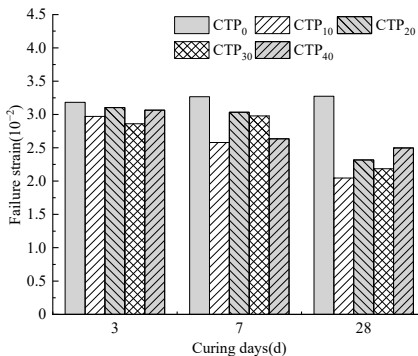

**Figure 8.** The failure strain of CTP at different curing ages.

Figure 9 shows the failure strain of CTA at different curing ages. It can be seen that the failure strain of the CTB specimen containing brick aggregate at the curing age of 7 d is much smaller than that at the curing ages of 3 d and 28 d. This may be because, at the beginning of the curing time of backfill, the hydration reaction of cement consumes free water generated during the specimen preparation. However, with the extension of the curing age, a large amount of free water is consumed along with the hydration reaction, and the inside of the backfill changes from a water saturation state to an unsaturated state. Then the absorption water in the aggregate is released to form free water and then reacts with the cement. Because of the high-water absorption of the brick aggregate, more water is absorbed around the particles. When the water is released, the cement and fine particles in the brick aggregate react first around the brick aggregate particles, resulting in greater friction and adhesion between the particles around the brick aggregate than the tailings particles. Under pressure, the friction force between the particles around the brick aggregate is greater than the sliding force between other particles, and the close connection between the brick aggregate and its surrounding particles is like a coarser particle. The skeleton structure between the coarser particles like this inhibits the movement of fine

particles inside the CTB. It prevents the compaction of some pores inside the CTB and the displacement restructuring between particles. So the failure strain is significantly reduced. With sufficient hydration reaction around the brick aggregate, water will be released outside the surrounding brick aggregate particles. Therefore, under a longer curing age, the hydration reaction and pozzolanic reaction inside CTB are more full and more uniform, the friction and adhesive force differences between all particles are weakened, and the pore compaction and displacement restructuring can be carried out normally under pressure.

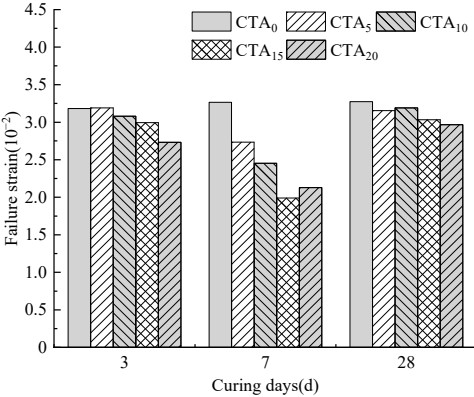

**Figure 9.** The failure strain of CTA at different curing ages.

Figure 10 shows the failure mode of some CTB specimens in an unconfined compression test. It can be found that $CTP_0$, with a curing age of 7 days, has S-type failure cracks and axial tensile cracks without showing obvious shear failure characteristics, as shown in Figure 10a. The $CTP_{10}$ with a curing age of 7 days (Figure 10b) shows tensile failure. The $CTA_{15}$, with a curing age of 7 days, exhibits a mixed failure of tension and shear, with dislocation and separation phenomena, as shown in Figure 10c. This is due to the development of internal cracks and imbalanced displacement recombination between particles in the CTB specimen under compression. From Figure 10d–f, it can be observed that $CTP_0$, $CTP_{10}$, and $CTA_{15}$ with a curing age of 28 days, all exhibit tensile failure, but $CTP_{10}$ exhibits more obvious brittle behavior.

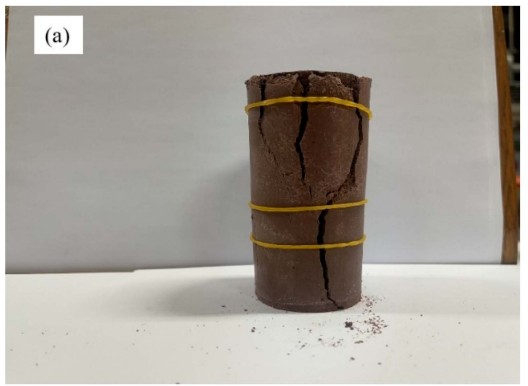
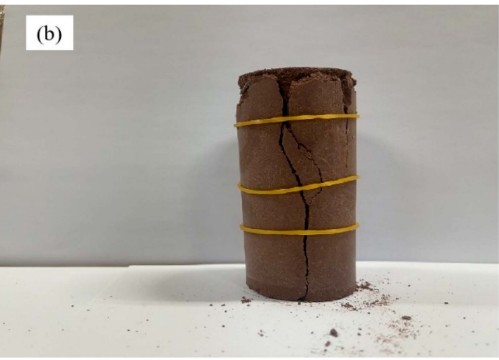

**Figure 10.** *Cont.*

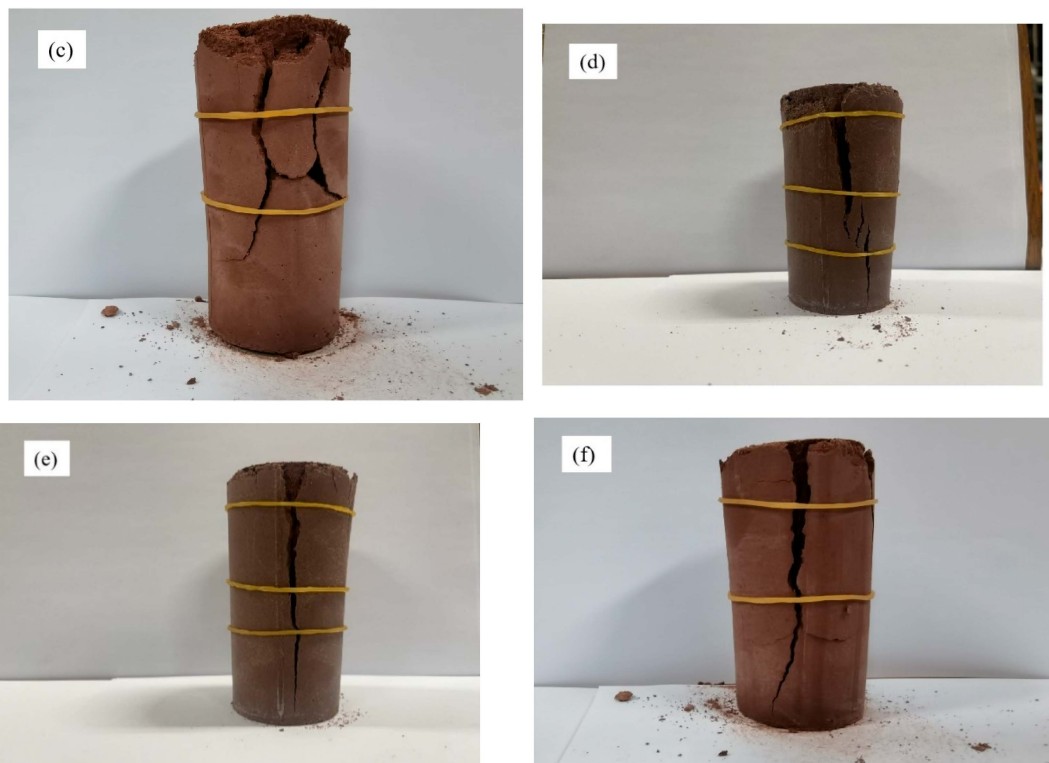

**Figure 10.** Failure mode: (**a**) $CTP_0$ for curing age at 7 d; (**b**) $CTP_{10}$ for curing age at 7 d; (**c**) $CTA_{15}$ for curing age at 7 d; (**d**) $CTP_0$ for curing age at 28 d; (**e**) $CTP_{10}$ for curing age at 28 d; (**f**) $CTA_{15}$ for curing age at 28 d.

*3.5. Microscopic Analysis*

Scanning the microstructure of brick powder and brick aggregate through electron microscopy, as shown in Figure 11a,b. It can be seen that during the crushing process, most of the brick powder particles are spherical and ellipsoidal, with uneven and rough surfaces. These spherical and ellipsoidal particles have a certain lubricating effect in the CTB slurry, which is also the reason for the small change in the slump of CTB-containing brick powder. It can also be seen that the brick aggregate particles have characteristics such as grooves, cracks, and cavities, and many fine particles are adsorbed on the brick aggregate. The grooves and cracks of brick aggregate particles increase their surface area in contact with water, and their cracks can accommodate more water. This is also the reason for the good water retention performance of brick aggregates. In addition, fine particles adsorbed on the brick aggregate will be more firmly connected around the brick aggregate particles than around the tailing particles after the pozzolanic reaction. This is also one of the reasons why the friction and bonding forces between the particles of CTB-containing brick aggregates are unbalanced. From Figure 11c, it can be seen that the distribution of C-S-H between particles is not uniform.

From (d,e), it can be noted that compared with CTP0, the internal structure seems denser, the C-S-H gel is more closely connected, and the number of AFt seems less. This also indicates that the increased density of the filling material containing brick powder is also the reason for its increased strength. In addition, it can be found from Figure 12a,b that ettringite (AFt) and gypsum were found in the XRD study of $CTP_0$, but two substances were not detected in $CTP_{20}$, which may be because the content of ettringite and gypsum is too small. Most sulfate ions that produce these two substances come from cement, so the formation of ettringite may be related to the content of cement.

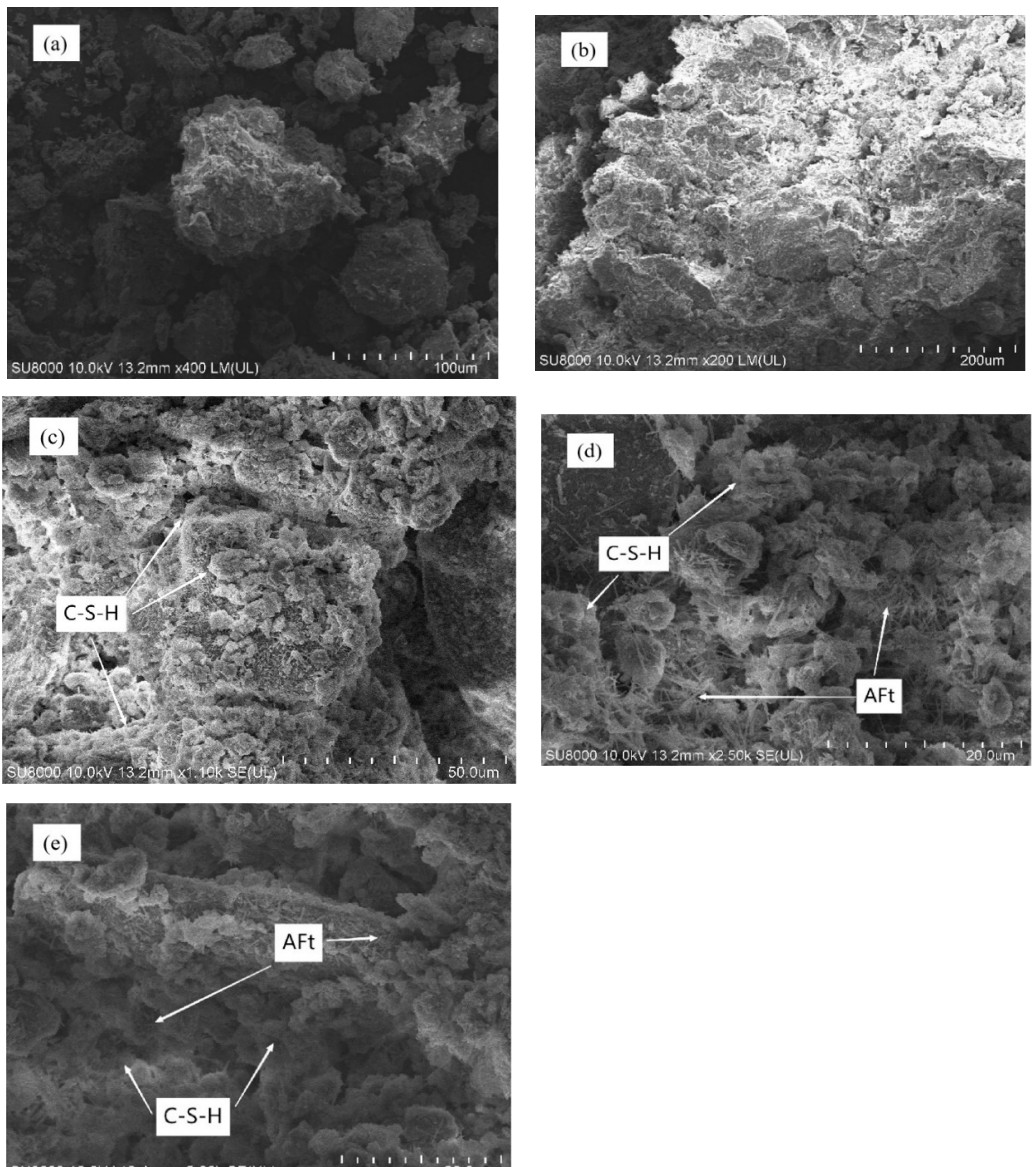

**Figure 11.** SEM images: (**a**) Brick powder sample; (**b**) Brick aggregate sample; (**c**) CTA$_{10}$ at 7 d of curing age; (**d**) CTP$_0$ at 28 d of curing age; (**e**) CTP$_{20}$ at 28 d of curing age.

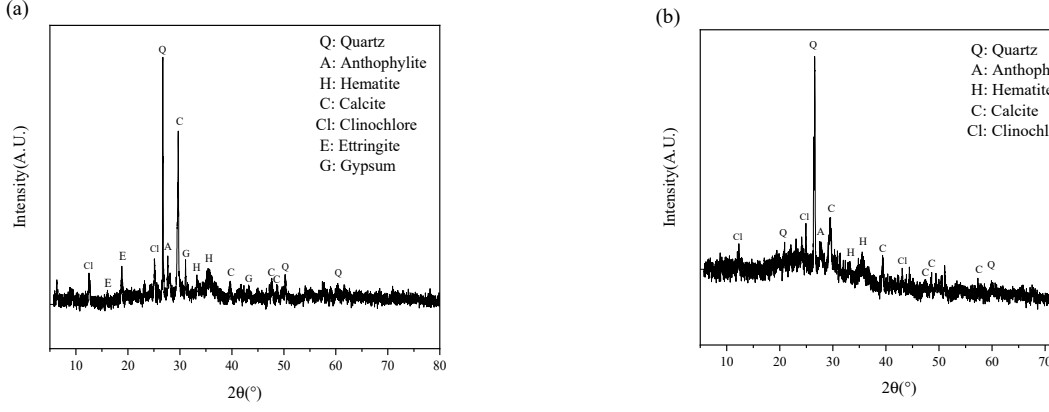

**Figure 12.** XRD pattern of CTB specimen with curing age of 28 d: (**a**) CTP$_0$; (**b**) CTP$_{20}$.

## 4. Establishment of Damage Constitutive Model

To visually represent the damage of brick powder backfill, a damage constitutive model was constructed in this paper to measure the damage degree of each specimen. According to the Lemaitre strain equivalence principle [48], the strain caused by the full stress applied to the damaged material should be equal to the strain caused by the effective force applied to the non-destructive material, which is Formula (1):

$$\varepsilon = \frac{\sigma}{E\prime} = \frac{\sigma\prime}{E} \tag{1}$$

According to continuum Damage mechanics [49], the CTB damage Constitutive equation can be expressed as Formula (2):

$$\sigma = (1 - D)E\varepsilon \tag{2}$$

Since the Weibull strength distribution function can fully reflect the influence of material defects and stress concentration sources on materials, this paper assumes that the mechanical properties of the studied CTB materials conform to the Weibull strength distribution [50] at the micro level and its probability density can be expressed as Formula (3) [51]:

$$p(F) = \frac{m}{F_0}\left(\frac{m}{F_0}\right)^{m-1}\exp\left(-\frac{F}{F_0}\right)^m, \tag{3}$$

where $F$ is Weibull distribution variable, $m$ and $F_0$ are Weibull distribution parameters.

Under a certain stress condition, the volume of the damaged unit in CTB is Formula (4):

$$V_P = \iiint_V \int_0^F p(F)dFdxdydz \tag{4}$$

Therefore, the damage variable can be defined as the ratio between the volume of destroyed units and the volume of total units, as shown in Formula (5):

$$D = \frac{V_P}{V} = 1 - \exp\left[-\left(\frac{F}{F_0}\right)^m\right] \tag{5}$$

Equation (5) is the damage evolution equation constructed by Weibull strength distribution. Since Strain energy can better explain the whole process of crack growth, this paper takes the strain variable as Weibull distribution, so the damage evolution equation of Equation (6) can be expressed as [52]:

$$D = 1 - \exp\left[-\left(\frac{\varepsilon}{F_0}\right)^m\right] \tag{6}$$

By substituting Formula (6), the damage constitutive model can be written as Formula (7):

$$\sigma = E_r\varepsilon\exp\left[-\left(\frac{\varepsilon}{F_0}\right)^m\right], \tag{7}$$

where $E_r$ is the elastic modulus of the damaged specimen.

According to the stress-strain curves of the lab test results, the boundary conditions can be obtained as Formula (8):

$$\begin{cases} \sigma|_{\varepsilon=0} = 0 \\ \sigma|_{\varepsilon=\varepsilon_{max}} = \sigma_{max} \\ \frac{d\sigma}{d\varepsilon}|_{\varepsilon=0} = E \\ \frac{d\sigma}{d\varepsilon}|_{\varepsilon=\varepsilon_{max}} = 0 \end{cases}, \tag{8}$$

where $\sigma_{max}$ is the peak stress; $\varepsilon_{max}$ is failure strain.

By substituting Formula (8) into the damage constitutive model, Formulas (9) and (10) of Weibull distribution parameters can be obtained:

$$F_0 = \frac{\varepsilon_{max}}{\left(\frac{1}{m}\right)^{\frac{1}{m}}} \tag{9}$$

$$m = \frac{1}{\ln \frac{E_r \varepsilon_{max}}{\sigma_{max}}} \tag{10}$$

By substituting Formulas (9) and (10) into Formulas (6) and (7), the damage evolution equation and damage constitutive model can be written as Formulas (11) and (12):

$$D = 1 - \exp\left[\ln \frac{\sigma_{max}}{E_r \varepsilon_{max}} \cdot \left(\frac{\varepsilon}{\varepsilon_{max}}\right)^{\frac{1}{\ln \frac{E_r \varepsilon_{max}}{\sigma_{max}}}}\right] \tag{11}$$

$$\sigma = E_r \varepsilon \exp\left[\ln \frac{\sigma_{max}}{E_r \varepsilon_{max}} \cdot \left(\frac{\varepsilon}{\varepsilon_{max}}\right)^{\frac{1}{\ln \frac{E_r \varepsilon_{max}}{\sigma_{max}}}}\right] \tag{12}$$

Formula (12) was fitted with the curve from the lab test results. It was found that the fitting results of CTB specimens containing brick powder and brick aggregate under different curing ages were relatively poor, and the differences were similar. Therefore, only CTB specimens containing brick powder at a curing age of 7 days were taken as examples, as shown in Figure 13.

It can be found that the Stress–strain curve of the test curve before the plastic deformation stage is concave upward, which is consistent with the stress-strain curve of CTB of iron-bearing tailings materials in other studies [53,54]. The theoretical curve is convex, indicating that this damage constitutive model cannot accurately simulate the specimen's initial compaction stage and linear elastic stage. In addition, it can be seen that the fitting effect of peak stress and the initial stage of post-peak failure is relatively good. Still, it does not accurately reflect the bearing capacity of the specimen after failure.

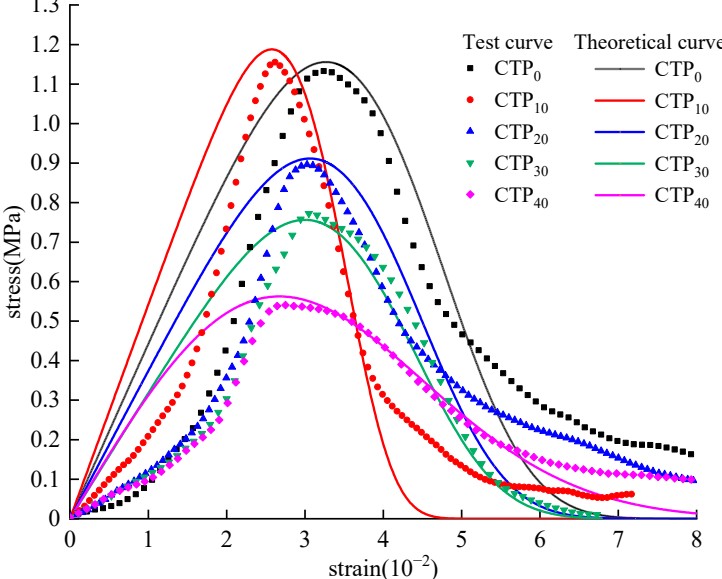

**Figure 13.** Fitting curve of damage constitutive model of CTP at the curing age of 7 d.

According to the test results, there is a nonlinear relationship between stress and strain at the initial compaction stage, and the curve is up-concave. According to the research of

domestic and foreign scholars [55], the relationship between stress and strain in the initial compaction stage can be expressed as Formula (13):

$$\sigma = \sigma_D \left(\frac{\varepsilon}{\varepsilon_D}\right)^2,\tag{13}$$

where $\sigma_D$ is the maximum stress at the initial compaction stage and $\varepsilon_D$ is the maximum strain in the initial compaction stage.

In the linear elastic stage, the stress-strain curve presents a shape approximated to a straight line. Therefore, assuming that no damage occurs to all specimens, Formula (14) is represented by a first-order function at the elastic stage:

$$\sigma = E(\varepsilon - \varepsilon_D) + \sigma_D,\tag{14}$$

where $E$ is the elastic modulus in the elastic stage.

To improve the fitting degree of the bearing capacity of backfill in the post-peak failure stage, referring to the study of CTB damage constitutive model containing different fibers by Wang et al. [56], the correction factor $\alpha$ was introduced to characterize the post-peak strength characteristics of backfill. Based on Formula (2), the damage constitutive relationship could be established as Formula (15):

$$\sigma = E_r(1 - \alpha D)\varepsilon\tag{15}$$

The damage constitutive model can be written as Formula (16):

$$\sigma = E_r\varepsilon\left\{1 - \alpha\left\{1 - \exp\left[-\left(\frac{\varepsilon}{F_0}\right)^m\right]\right\}\right\}\tag{16}$$

By substituting the boundary conditions of Formula (8) into Formula (16), Weibull distribution parameters can be obtained by Formulas (17) and (18):

$$F_0 = \left(\frac{m\varepsilon_{\max}^m}{\frac{\sigma_{\max}}{\sigma_{\max} + E_r\varepsilon(1-\alpha)}}\right)^{\frac{1}{m}}\tag{17}$$

$$m = -\frac{\frac{\sigma_{\max}}{\sigma_{\max} + E_r\varepsilon_{\max}(\alpha-1)}}{\ln\frac{\sigma_{\max} + E_r\varepsilon_{\max}(\alpha-1)}{E_r\varepsilon_{\max}\alpha}}\tag{18}$$

To simplify the calculation, let $\lambda = \sigma_{\max}/[\sigma_{\max} + E_r\varepsilon_{\max}(\alpha - 1)]$ and keep $m$ as the fitting parameter of the damage constitutive model. By substituting Formulas (17) and (18) into Formula (16), the simplified damage evolution equation and damage constitutive model can be written as Formulas (19) and (20):

$$D = 1 - \exp\left[-\frac{\lambda}{m} \cdot \left(\frac{\varepsilon}{\varepsilon_{\max}}\right)^m\right]\tag{19}$$

$$\sigma = E_r\varepsilon\left\{1 - \alpha\left\{1 - \exp\left[-\frac{\lambda}{m} \cdot \left(\frac{\varepsilon}{\varepsilon_{\max}}\right)^m\right]\right\}\right\}\tag{20}$$

Combining Formulas (13), (14) and (20), the three-stage damage constitutive model of CTB is obtained, as shown in Formula (21):

$$\sigma = \left\{\begin{array}{ll} \sigma_D\left(\frac{\varepsilon}{\varepsilon_D}\right)^2 & \sigma < \sigma_D \\ E(\varepsilon - \varepsilon_D) + \sigma_D & \sigma_D \leq \sigma < \sigma_P \\ E_r\varepsilon\left\{1 - \alpha\left\{1 - \exp\left[-\frac{\lambda}{m} \cdot \left(\frac{\varepsilon}{\varepsilon_{\max}}\right)^m\right]\right\}\right\} & \sigma_P \leq \sigma \end{array}\right\},\tag{21}$$

where $\sigma_P$ is the yield point stress.

According to the concept of threshold point stress proposed by Martin et al. [57] and combined with the test curve of each specimen, it is found that the threshold point will change accordingly because the internal structure distribution of CTB is not the same. According to the actual fitting effect, the threshold point of the maximum stress in the initial compaction stage of the CTB specimen is shown in Formula (22):

$$\sigma_D = 30\%\sigma_{\max} \tag{22}$$

Similarly, the threshold point of yield point stress of different CTB specimens is shown in Equation (23):

$$\sigma_P = (90\% \sim 95\%\sigma_{\max}) \tag{23}$$

## 5. Verification and Analysis of the Model

Peak stress, failure strain, elastic modulus, maximum stress at the initial compaction stage and yield point stress of CTB can be obtained by the stress-strain curve from the lab test results. The model curves and damage constitutive equations of CTP and CPA can be obtained by calculating the elastic modulus of the damaged specimen under each curve $E_r$ and introducing the correction factor $\alpha$. The three-stage theoretical curve and test curve of CTB are shown in Figure 14. It can be seen that the theoretical curve is in good agreement with the test curve, especially the initial compaction stage, the linear elastic stage, and the post-peak failure stage after 90% of the peak stress. This indicates that the correction factor $\alpha$ has a good effect on correcting the post-peak curve. In addition, it can be found that some theoretical and test curves have certain differences in the plastic deformation stage and the initial stage of the post-peak failure stage because the post-peak curve drop rate is too high. It can also be seen from the experimental curve that the initial compaction stage of CTB is too long, which indicates that there are many pores and micro-cracks inside CTB, which will lead to certain randomness in the compression process.

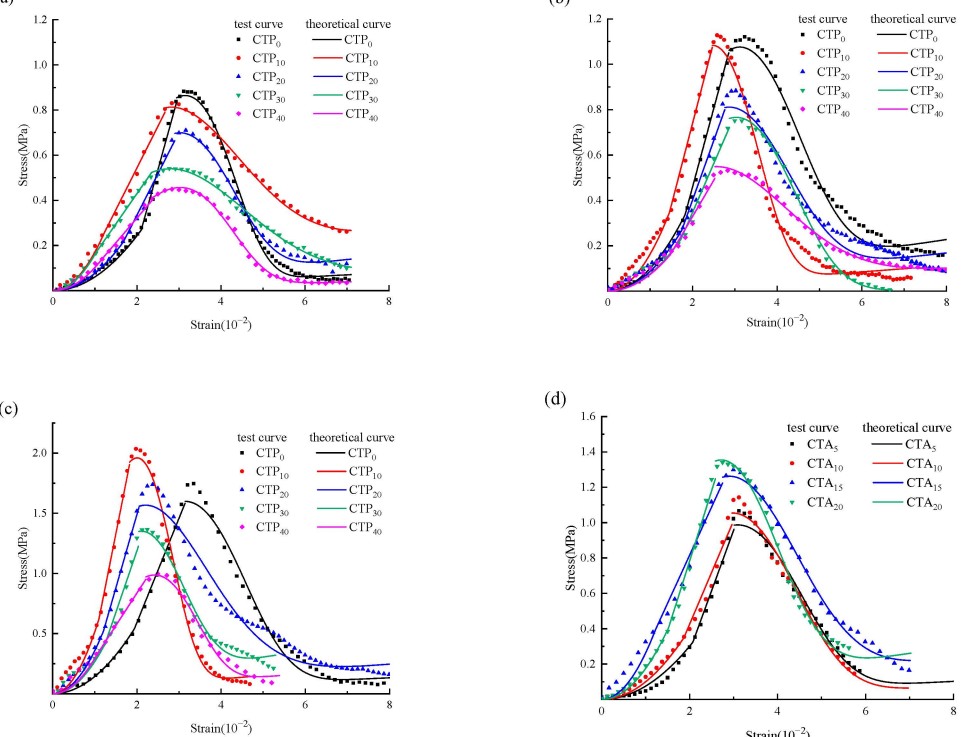

**Figure 14.** *Cont.*

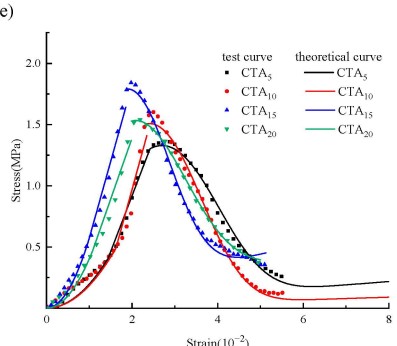
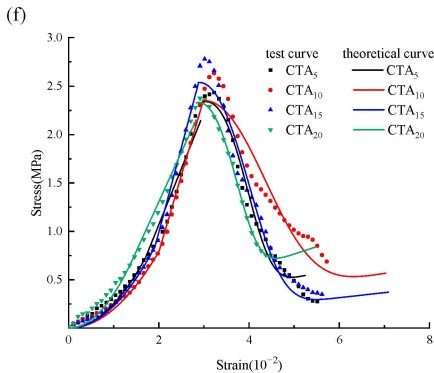

**Figure 14.** Comparison of theoretical curves and test curves: (**a**) CTP for curing age at 3 d; (**b**) CTP for curing age at 7 d; (**c**) CTP for curing age at 28 d; (**d**) CTA for curing age at 3 d; (**e**) CTA for curing age at 7 d; (**f**) CTA for curing age at 28 d.

## 6. Conclusions

In this study, the CTB was prepared after the partial replacement of cement with brick powder and the partial replacement of tailings with brick aggregates. The effects of brick powder and brick aggregate on the bleeding rate, slump value, UCS and failure strain of CTB were investigated. Based on the lab test results, the stress-strain curves and damage processes of CTB specimens were investigated, and a damage constitutive model of the CTB specimens was established. The main findings of this study are as follows:

1. CTB containing brick powder increases in bleeding rate and decreases in slump value as the content of replacement cement increases. The CTB containing brick aggregate decreases in bleeding rate and slump value as the content of replacement tailings increases. Considering that the slump far meets the requirements of backfill standards and the high bleeding rate, both CTB materials can be added with an appropriate amount of thickener or increased concentration.

2. The CTB with brick powder is lower in strength than the original CTB in the early stages and changes in strength in the later stages. The UCS of CTB with a 10% brick powder content is highest at 7 and 28 d of curing age, and at 28 d of curing age, the UCS of CTB with a 20% brick powder content can be essentially the same as the original CTB. CTB up to 20% brick aggregate content is favourable for UCS at all ages, with the highest strength of the CTB at 20% brick aggregate content at curing age 3 d and the highest strength of the CTB at 15% brick aggregate content at curing ages 7 d and 28 d. If for better cost-effectiveness, it is recommended to use brick powder with a content of less than 20% to replace cement. In addition, it is recommended to use 15% brick aggregate instead of tailings for higher strength.

3. CTB-containing brick powder causes a decrease in peak strain in unconfined compression due to its dense nature, the hydration products generated by the pozzolanic reaction and the AFt content, which is particularly noticeable at the curing age of 28 d. In contrast, at 7 d of curing age, CTB with brick aggregate causes a significant drop in peak strain due to an imbalance in the reaction around the brick aggregate particles and the tailings particles. The lower the failure strain, the more stable the backfill material is under low stress. However, considering the brittle behavior of CTB containing brick powder in the later curing stage, fiber materials can be added to the CTB material production process to cope with the underground environment under high strain. In addition, CTB containing brick aggregates should reduce the use of limit states within the curing period of 28 days.

4. The effect of coarse aggregate Weibull strength distribution function is used to establish a damage constitutive model. The ordinary damage constitutive model fits the peak stress of the specimen relatively well but does not consider the problem that the specimen still has a bearing capacity after failure. At the same time, it is found that none of the fitting results can effectively fit the initial compaction stage and linear

elastic stage of the specimen. The concept of threshold point is introduced, and the three parts of the compaction stage, the linear elastic stage and the stage after the linear elastic stage are fitted respectively by the piecewise method. This method solves the problem of poor fitting between the initial compaction stage and the linear elastic stage. It improves the fitting degree of damage constitutive model with a correction factor to the post-peak bearing capacity.

5. In conclusion, the three-stage damage constitutive model has a good effect on the characterization of damage of CTP and CTA. This provides an essential reference for predicting the stress and strain of filling materials with longer compression and linear elastic stages or filling materials containing iron ore tailings. It also helps to evaluate their performance changes under long-term loads and complex environmental conditions.

**Author Contributions:** Conceptualization, T.S. and Y.W.; methodology, Z.Y. and Y.Z.; validation, K.W. and T.S.; writing—original draft preparation, T.S. writing—review and editing, K.W. All authors have read and agreed to the published version of the manuscript.

**Funding:** This research was funded by the National Natural Science Foundation Project of China (Grant No. 301184).

**Data Availability Statement:** The datasets used or analyzed during the current study are available from the corresponding author upon reasonable request.

**Acknowledgments:** Thanks for the great efforts of editors and reviewers.

**Conflicts of Interest:** The authors declare no conflict of interest.

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
