# Peer review of "Effect of Waste Clay Bricks on the Performance of Cemented Tailings Backfill and Its Damage Constitutive Model"

_minerals, doi:10.3390/min13070987_

Round 1

Reviewer 1 Report

The article is interesting and relevant, devoted to the use of waste in the production of building ceramics. But, judging by the text of the article, the main part of the resulting ceramics is the tailings of iron ore production, which was not reflected either in the title of the article or in the text.

The Introduction does not consider various cases of using various wastes, including overburden, generated during mining, in the production of building ceramics.

L. 94-96 - the characteristics of raw materials and their origin are insufficient and incomprehensible. There are no photos of the original raw materials.

L. 114-115 - there is not enough information about where, how and by whom the analyzes were performed. And the mineral composition of raw materials has not been studied at all.

In the list of used literature there are many publications of the authors of the article, and in general there are too many articles written by authors from China. Do not forget that you are writing for an international journal for the global scientific community.

Reviewer 2 Report

There are several editorial and technical modifications that the authors should consider and implement before final acceptance of the manuscript.

- how does the microstructure analysis with the SEM compare to the physical lab test results?

- why is there such a large volume of waste bricks and do they originate directly from production (faulty casting) or construction demolition?

- what is the need of a complex theoretical formulation that seems to be out of place in an otherwise purely experimental study?

Editorial:

- Significant discussions are made with respect to the microstructure of the specimen in sections 3.3 and 3.4. However, this should be a section on its own where Fig. 9 is included. Microstructure is an important parameter that was analyzed in the research and more should be said about it.

- More literature review should be included with respect to pozzolans and other material being used in CPB. There are numerous references where components such as fly ash, slag, rice husk ash, and others are used as a component in CPB. After this general review, the topic of waste clay bricks can then be introduced.

- WCB should be identified as waste clay bricks at its first mention on line 55 and not later on lines 73-74.

Technical:

- The first technical clarification that should be expanded upon is the issue of waste clay bricks. It should be explained what is the volume of waste bricks rather than total brick production (lines 57-58), and it should be mentioned why 50-70% of construction waste is made up of bricks. Are these unused/faulty ones or those that come from demolition? Also, why would most of the 20 billion m3 of bricks end up as waste?

- Detailed analysis of SEM should be made in a separate section after 3.4. It should preferably be combined with x-ray diffraction analysis if possible to determine the new crystal phases formed. This part should shed more light on the slump, bleed, UCS, and strain results.

- Section 4 is a confusing addition to an otherwise classical test and analysis study on a CPB component. It is unclear why a theoretical damage constitutive model is required or how it would be used in the future. This is especially true as lines 327-331 indicate that a first comparison with the experimental data did not produce any fits. The authors end section 4 with a different formulation, which is then verified in section 5. Still, it is unclear why this model was developed and how useful it will be in the future. If a regular series of tests conducted on WCB was enough to provide an understanding of their performance in CPB, how would the theoretical model help future researchers?

The English language is very good but a final spelling and grammatical check should be made before publication.

Reviewer 3 Report

This is a good paper on a development and identification of a damage model parameters. Further on, the model can be implemented into a finite element program and used for solving the failure problems of the described material. 

I see one disadvantage. The literature review should be extended concerning the damage models. For example, there is a lack of reference concerning the fundamental Formula (2). This is the Kachanov's model.

https://link.springer.com/book/10.1007/978-94-017-1957-5

LM Kachanov, Introduction to continuum damage mechanics.

Springer1986,  Mechanics of Elestic Stability (MEST, vol 10)

Please be as kind as to improve the paper adding a wider literature review than it has been already given.

Author Response

I am very sorry that the order of reply was wrong due to my carelessness. Please see the attachment of this version I am sending now. I am here again to apologize to you for my carelessness. Wish you a happy day. Please see this attachment.

Reviewer 4 Report

The authors used waste clay bricks to invesitage the cemented paste backfill. The paper is generally good but it needs improvement. Followings should be carried out before acceptance:

The abstract should contain important results of the study.

How this recycled materials for this study is obtained?

Novelty is not clear. Very same studies are already exists. What is the difference?

The reason for selecting design mixture should be added.

Compare your results with existing studies

What is the constant slump value fo samples with 0-20

Other types of powder can also be used as cementiotous materials such as glass powder and coal bvottom ash. For this purpose the following studies also should be added: mechanical behavior in terms of shear and bending performance of reinforced concrete beam using waste fire clay as replacement of aggregate; Data-driven based estimation of waste-derived ceramic concrete from experimental results with its environmental assessment; influence of replacing cement with waste glass on mechanical properties of concrete; use of recycled coal bottom ash in reinforced concrete beams as replacement for aggregate; concrete containing waste glass as an environmentally friendly aggregate: a review on fresh and mechanical characteristics; mechanical behavior of crushed waste glass as replacement of aggregates;flexural behavior of reinforced concrete beams using waste marble powder towards application of sustainable concrete; Use of waste glass powder toward more sustainable geopolymer concrete; the use of Crushed Recycled Glass for Alkali Activated Fly Ash Based Geopolymer Concrete and Prediction of Its Capacity

Add photos for test setup?

The authors should verify their results with other studies. This will significanlty improve the paper.

The importance of recycling materials to overcome enviermental prbolem should be added to introduction using:  composition component influence on concrete properties with the additive of rubber tree seed shells; normal-weight concrete with improved stress–strain characteristics reinforced with dispersed coconut fibers; investigation of the physical-mechanical properties and durability of high-strength concrete with recycled pet as a partial replacement for fine aggregates; effects of waste powder, fine and coarse marble agregates on concrete compressive strength; improving bond performance of ribbed steel bars embedded in recycled aggregate concrete using steel mesh fabric confinement; production of perlite-based-aerated geopolymer using hydrogen peroxide as eco-friendly material for energy-efficient buildings; 

Please add damaged photos damaged photos of samples

Add recent studies on this subject to introduction. There are many studies on the introduction for this topic.

Conclusion should be improved. The recommendation consdiering all test should be given for engineers.

Round 2

Reviewer 4 Report

The paper can be accepted 
